# Alterations in Th17 Cells and Non-Classical Monocytes as a Signature of Subclinical Coronary Artery Atherosclerosis during ART-Treated HIV-1 Infection

**DOI:** 10.3390/cells13020157

**Published:** 2024-01-15

**Authors:** Tomas Raul Wiche Salinas, Yuwei Zhang, Annie Gosselin, Natalia Fonseca Rosario, Mohamed El-Far, Ali Filali-Mouhim, Jean-Pierre Routy, Carl Chartrand-Lefebvre, Alan L. Landay, Madeleine Durand, Cécile L. Tremblay, Petronela Ancuta

**Affiliations:** 1Département de Microbiologie, Infectiologie et Immunologie, Faculté de Médecine, Université de Montréal (UdeM), Montreal, QC H2X 0A9, Canada; tomas.raul.wiche.salinas@umontreal.ca (T.R.W.S.); zhangyw927@gmail.com (Y.Z.); c.tremblay@umontreal.ca (C.L.T.); 2CRCHUM, Montreal, QC H2X 0A2, Canada; annie.gosselin.chum@gmail.com (A.G.); nataliarosario@id.uff.br (N.F.R.); mohamed.el.far.chum@ssss.gouv.qc.ca (M.E.-F.); alifilali2009@gmail.com (A.F.-M.); carl.chartrand-lefebvre@umontreal.ca (C.C.-L.); madeleine.durand@umontreal.ca (M.D.); 3Chronic Viral Illness Service and Division of Hematology, Research Institute of the McGill University Health Centre, Montreal, QC H4A 3J1, Canada; jean-pierre.routy@mcgill.ca; 4Département de Radiologie, Radio-Oncologie et Médecine Nucléaire, Faculté de Médecine, Université de Montréal (UdeM), Montreal, QC H2X 0A9, Canada; 5Rush University Medical Center, Chicago, IL 60612, USA; alan_landay@rush.edu; 6Département de Médecine, Faculté de Médecine, Université de Montréal (UdeM), Montreal, QC H2X 0A9, Canada

**Keywords:** HIV-1, antiretroviral therapy (ART), cardiovascular disease (CVD), Th17/Treg cells, non-classical monocytes, myeloid/plasmacytoid dendritic cells

## Abstract

Cardiovascular disease (CVD) remains an important comorbidity in people living with HIV-1 (PLWH) receiving antiretroviral therapy (ART). Our previous studies performed in the Canadian HIV/Aging Cohort Study (CHACS) (>40 years-old; Framingham Risk Score (FRS) > 5%) revealed a 2–3-fold increase in non-calcified coronary artery atherosclerosis (CAA) plaque burden, measured by computed tomography angiography scan (CTAScan) as the total (TPV) and low attenuated plaque volume (LAPV), in ART-treated PLWH (HIV+) versus uninfected controls (HIV−). In an effort to identify novel correlates of subclinical CAA, markers of intestinal damage (sCD14, LBP, FABP2); cell trafficking/inflammation (CCL20, CX3CL1, MIF, CCL25); subsets of Th17-polarized and regulatory (Tregs) CD4^+^ T-cells, classical/intermediate/non-classical monocytes, and myeloid/plasmacytoid dendritic cells were studied in relationship with HIV and TPV/LAPV status. The TPV detection/values coincided with higher plasma sCD14, FABP2, CCL20, MIF, CX3CL1, and triglyceride levels; lower Th17/Treg ratios; and classical monocyte expansion. Among HIV^+^, TPV^+^ versus TPV^−^ exhibited lower Th17 frequencies, reduced Th17/Treg ratios, higher frequencies of non-classical CCR9^low^HLADR^high^ monocytes, and increased plasma fibrinogen levels. Finally, Th17/Treg ratios and non-classical CCR9^low^HLADR^high^ monocyte frequencies remained associated with TPV/LAPV after adjusting for FRS and HIV/ART duration in a logistic regression model. These findings point to Th17 paucity and non-classical monocyte abundance as novel immunological correlates of subclinical CAA that may fuel the CVD risk in ART-treated PLWH.

## 1. Introduction

Mortality in people living with human immunodeficiency virus type 1 (HIV) (PLWH) has considerably diminished after the implementation of antiretroviral therapy (ART). Nevertheless, non-AIDS comorbidities, such cardiovascular disease (CVD), remain highly prevalent in ART-treated PLWH [1,2,3,4,5,6,7,8]. A modeling study performed in the Netherlands estimated an increase in the life expectancy of ART-treated PLWH from 43.9 years in 2010 to 56.6 years in 2030 and predicted that 78% of PLWH would be diagnosed with CVD [9,10]. Indeed, PLWH tend to present clinical signs of CVD approximately 10 years earlier compared to the general population [11]. In addition to traditional CVD risk factors (e.g., smoking, hypertension, dyslipidemia, diabetes, insulin resistance, and/or sedentary life-style) [12] and mental health disorders with an impact on CVD risk, HIV-specific mechanisms (e.g., HIV-mediated metabolic alterations due to long-term administration of ART) contribute to the premature occurrence of CVD in ART-treated PLWH [7,8,10,13,14,15]. In addition, the persistence of HIV reservoirs during ART is associated with impaired intestinal mucosal barrier function, microbial translocation, immune dysfunction, and chronic inflammation, which, together with CMV and other co-infections, contribute to an increased CVD risk in this group [16,17,18,19,20]. Thus, CVD represents a major non-AIDS comorbidity in ART-treated PLWH and novel therapeutic interventions are needed to reduce this risk.

Cells of the adaptive immune system are key players in CVD pathogenesis [12,21,22]. Among them, T-lymphocytes infiltrate atherosclerotic plaques and are recruited into the heart via mechanisms involving the hepatocyte growth factor receptor c-Met, CCR4, CXCR3, and CCR5 [23]. The antigenic specificity of such CD4^+^ T-cell subsets remains yet unclear [24]. In HIV-uninfected individuals, T-helper 1 (Th1) cells have pro-atherogenic features, whereas regulatory T-cells (Tregs) and Th17 cells are reported to play dual roles in the development of atherosclerosis, either protective or pathogenic [24,25]. In individuals with unstable angina or acute myocardial infarction, the frequency of Th17 cells was higher compared to those in participants with stable angina or healthy individuals [26]. The role of CD4^+^ T-cells and their associated cytokines in atherosclerosis or coronary artery disease in ART-treated PLWH remains poorly documented, with some recent studies documenting the expansion of Tregs [27] and surprisingly Th17 cells [28]. Moreover, ART-treated PLWH with carotid atherosclerotic plaques, increased carotid intima media thickness, or arterial stiffness presented with increased circulating CD8^+^ T-cell activation (CD38^+^HLA-DR) [29,30].

In addition to T-cells, innate immune cells, such as monocytes and myeloid dendritic cells (mDCs), also contribute to atherosclerotic plaque formation/rupture [31,32,33]. Mice deficient in CCR2 and CX3CR1, two chemokine receptors important for the migration of classical (CD14^++^CD16^−^), intermediate (CD14^++^CD16^++^), and non-classical (CD14^dim^CD16^++^) monocytes, respectively, exhibited decreased atherosclerotic plaque severity, thus pointing to the deleterious role of these monocytes in CVD [34,35]. Studies in mice deficient in Nur77/NR4A1, which lacked non-classical monocytes, reported controversial results that may be explained by differences in study design [35]. Nevertheless, human studies support a deleterious role of monocytes in CVD pathogenesis [35], as well as the pro-inflammatory features of non-classical monocytes [36]. In PLWH compared to uninfected individuals, arterial inflammation is higher and correlates with increased circulating levels of the pro-inflammatory cytokine IL-6 and monocyte activation [37,38]. In uninfected individuals, an elevated frequency of circulating non-classical monocytes was associated with an amplified carotid intima-media thickness (IMT) over 10 years in a prospective cohort study [39]. Similarly, in PLWH, the intermediate monocyte counts were associated with subclinical atherosclerosis [40] and the expression of CX3CR1 on CD16^+^ monocytes independently predicted carotid artery thickness [41]. Of note, the decreased expression of CXCR4 was observed on non-classical monocytes in women with subclinical atherosclerosis [42]. Moreover, ex vivo experiments with monocytes of virologically suppressed PLWH showed increased potential to form atherosclerosis-promoting foam cells compared to uninfected individuals [43]. Furthermore, the atherosclerosis plaque burden was associated with increased levels of monocyte chemoattractant protein-1 (MCP-1/CCL2) in ART-treated PLWH [44,45].

Other innate immune cells, such as plasmacytoid DCs (pDCs), are documented to infiltrate atherosclerotic lesions [46,47], but their contribution to atherosclerosis remains controversial. Some mice studies have shown that pDCs promote atherogenesis through their capacity to produce interferon (IFN)-α [48], while others support their role in atheroprotection [49]. pDCs contribute to peripheral and central tolerance by the induction of Tregs; meanwhile, the anti-atherogenic role of pDCs was associated with the indoleamine 2,3-dioxygenase 1 (IDO1)-dependent induction of aortic Treg cells [50]. During HIV infection, there is a decline in pDC counts and alterations in their function, which are not restored with ART [51,52]. Whether interventions to restore pDCs may limit the CVD risk in ART-treated PLWH requires investigation.

The objective of this study was to explore the existence of an immunological signature associated with subclinical coronary artery atherosclerosis (CAA) in ART-treated PLWH in local Canadian cohorts. To reach this objective, we had access to plasma and PBMC samples collected at baseline from ART-treated PLWH (HIV^+^; *n* = 61) and HIV-uninfected participants (HIV^−^; *n* = 21) included in the cardiovascular imaging sub-study of the Canadian HIV/Aging Cohort Study (CHACS), a multi-center prospective controlled cohort study initiated in 2011 [53,54]. Our results revealed alterations in Th17/Treg ratios and the frequencies of non-classical CCR9^low^HLA-DR^high^ monocytes, together with overt plasma fibrinogen levels, which all correlated with the magnitude of subclinical CAA in ART-treated PLWH. Of particular importance, the predictive CVD risk value of Th17/Treg ratios and non-classical CCR9^low^HLA-DR^high^ monocyte expansion remained valid upon adjustment to the traditional CVD risk factors included in the Framingham risk score (FRS) (i.e., smoking, statins, LDL, triglycerides), as well as HIV-specific parameters (HIV/ART duration). Our findings emphasize the need for novel therapies aimed at restoring Th17 paucity at the mucosal level and reducing monocyte-mediated inflammation to limit the CVD risk in ART-treated PLWH.

## 2. Materials and Methods

### 2.1. Study Participants

The design of the Canadian HIV and Aging Cohort Study (CHACS) has been previously reported [53,54,55]. The first eligible HIV^−^ (*n* = 21) and HIV^+^ (*n* = 61) participants recruited in the CHACS were included in this sub-study. Briefly, the inclusion criteria for the cardiovascular imaging sub-study of the CHACS cohort were subjects older than 40 years old, without clinical manifestations or a diagnosis of CAA at recruitment, and with a 10-year risk of cardiovascular disease according to the Framingham risk score (FRS, calculated based on age, HDL and cholesterol levels, systolic blood pressure, smoking, diabetes, statin treatment [53,54,55]) ranging from 5 to 20% (Table 1). Exclusion criteria were renal impairment and hypersensitivity to contrast agents. Plasma and PBMC samples collected at baseline from ART-treated PLWH (HIV^+^) and uninfected (HIV^−^) participants were available for this study.

### 2.2. Computed Tomography Angiography Scan

All study participants underwent a computed tomography angiography scan (CTA scan) to determine the total plaque volume (TPV) and low attenuated plaque volume (LAPV, high-risk atherosclerotic plaque rupture), as previously described [5,55]. Briefly, TPV represents the total volume (calculated from 3D reconstructions) of all coronary artery plaques present in an individual’s coronary arteries. Coronary arteries that are normal (free of signs of atherosclerosis) will have a TPV and LAPV of zero, while any volume greater than zero indicates the presence of CVD. For this sub-study, HIV^+^ and HIV^−^ participants were stratified based on TPV and LAPV values of zero or greater than zero, indicative of the absence or the presence of CVD, respectively (Table 2).

### 2.3. ELISA

Markers of microbial translocation (LPS binding protein, LBP), mucosal damage (intestinal-type fatty acid biding protein, I-FABP), immune activation (sCD14), and chemokines (CCL20, CX3CL1, CCL25, MIF) were quantified in plasma by ELISA, according to the manufacturer’s protocol (R&D Systems, Minneapolis, MN, USA).

### 2.4. Flow Cytometry

Fluorescence-activated cell sorting (FACS) was used to identify subsets of CD3^+^CD4^+^ T-cells (i.e., Th17 (CCR6^+^CD26^+^CD161^+^), Tregs (CD127^−^CD25^+^Foxp3^+^), central memory (CM, CD45RA^−^CCR7^+^), effector memory (EM, CD45RA^−^CCR7^−^), effector memory RA^+^ (TEMRA, CD45RA^+^CCR7^−^), and naive (CD45RA^+^CCR7^+^) cells); subsets of monocytes (CD3^−^CD4^low^HLA-DR^+^CD1c^−^) (i.e., classical (CD14^++^CD16^−^), intermediate (CD14^++^CD16^+^), non-classical (CD14^−^CD16^++^), and Slan/M-DC8^+^); and myeloid (mDC; HLA-DR^+^CD1c^+^) and plasmacytoid dendritic cells (pDC, BDCA2^+^CD123^+^) and measure their expression of chemokine receptors (i.e., CCR2, CCR6, CCR9, and CX3CR1). Cells were analyzed by FACS using a BD-LSRII cytometer, BD-Diva (BD Biosciences, Franklin Lakes, NJ, USA), and FlowJo version 10 (Tree Star, Inc., Ashland, OR, USA). The antibodies used in this study are presented in Appendix A. All Abs were titrated for an optimal noise/ratio. The gating strategy used to identify the different immune cell subsets is described in Appendix A. A combination of gates using the Boolean tool gating from FlowJo was used to determine the expression of chemokines. All results generated by flow cytometry are included in Appendix A.

### 2.5. Statistical Analysis

The Shapiro–Wilk test was used to determine the normal distribution of the continuous variables. Variables with a normal distribution were presented as means with standard deviation, and variables with a non-normal distribution were presented as the median with interquartile range (IQR). The comparison of two groups for the same variable with a normal distribution was analyzed using a *t*-test, and non-normally distributed variables were analyzed with Mann–Whitney. Categorical variables were represented as proportions, and comparisons for the same variable between groups were analyzed using Fisher’s exact test. To assess the effect of potential confounding factors on the association between the immune subset and the presence or absence of coronary plaques, logistic regression analyses were implemented in R (R Core Team R: A Language and Environment for Statistical Computing Vienna, Austria, 2019; Available from: https://www.R-project.org) [56]. First, we compared univariate and multivariate logistic regression models to study the contribution of the immune subset alone and in the presence of four sets of potential confounding factors (Models 1–4), as well as fibrinogen alone (Table 3). Model 1 adjusted for the ART duration, smoking, and the use of statins. Model 2 adjusted for the ART duration, smoking, and LDL. Model 3 adjusted for the ART duration, smoking, and triglycerides. Model 4 adjusted for the ART duration, HIV duration, and FRS. The confounding effect was estimated by the measure of the association (OR, odds ratio) before and after adjusting (% Δ (delta) OR). A change in the estimated association measure of less than 10% was used to exclude confounders, while a change superior to 30% identified large confounding bias [56]. *p*-value adjustment for the multiple testing hypothesis was performed according to the method of Benjamin and Hochberg [57], which controls the false discovery rate with adjusted *p*-value cutoffs of 0.05.

## 3. Results

### 3.1. Laboratory and Imaging Markers of Subclinical Atherosclerosis in ART-Treated PLWH

In previous studies performed by our group, the CTA scan was used as a noninvasive tool to visualize and quantify CAA plaques as the total (TPV) and low attenuated plaque volume (LAPV) in the ART-treated PLWH (HIV^+^) and uninfected controls (HIV^−^) included in our CHACS cohort [5,54,55]. In the current cardiovascular imaging sub-study, a total of *n* = 61 HIV^+^ and *n* = 21 HIV^−^ were included, with the demographic and clinical characteristics depicted in Table 1. Briefly, differences between the HIV^+^ and HIV^−^ groups in terms of age (mean 55.38 versus 56.94 years), body mass index (BMI; median 24.43 versus 25.72 kg/m^2^), and FRS (median 10 versus 8) did not reach statistical significance; also, similar proportions of participants received treatment with statins (23% versus 23.8%) (Table 1). The HIV^+^ group included 61/61 males (100%), while the HIV^−^ group included only 17/21 males (81%; *p* = 0.003) (Table 1). The measurement of the CAA plaque volume using CT scan angiograms was performed in all sub-study participants, as previously reported [5,55]. The CAA plaque prevalence was similar between the HIV^+^ and HIV^−^ groups (Table 1). Indeed, among the HIV^+^ group, 39/61 (64%) participants were identified with detectable plaques (>1), while 22/61 (36%) participants had undetectable plaques (<1) measured as TPV and LAPV (plaque volume expressed in mm^3^) (Appendix A; Table 1). The HIV^−^ group included 13/21 (62%) with detectable plaques, while 8/21 (38%) participants had undetectable plaques (Appendix A; Table 1). Similarly, differences in the CAA plaque volume between the HIV^+^ and HIV^−^ groups did not reach statistical significance (TPV (median 108.3 versus 49.7) and LAPV values (median 34.4 versus 10.53); *p* = 0.32 and 0.39, respectively) (Table 1).

In terms of laboratory parameters, the HIV^+^ and HIV^−^ groups presented with similar white blood counts, lymphocyte counts, and LDL levels; however, the HIV^+^ compared to the HIV^−^ group had significantly lower levels of HDL (median: 1.23 versus 1.44) and higher levels of triglycerides (median: 1.73 versus 1.21) (Table 1). Since low HDL and high triglycerides are two well-documented CVD risk factors [58], such differences may accelerate the occurrence of subclinical CVD events in ART-treated PLWH, consistent with our recently reported findings [5].

We further sought to identify clinical and laboratory markers associated with subclinical CAA plaques among HIV^+^ participants (Table 2). Considering the fact that the TPV and LAPV values were strongly positively correlated in HIV^+^ and HIV^−^ participants (Appendix A; *p* < 0.0001, r = 0.9913), all subsequent statistical analyses were based on the TPV values equal to or higher than zero. In terms of demographics and clinical parameters, TPV^+^ and TPV^−^ HIV^+^ participants were similar in age (mean: 56.24 versus 53.85 years), sex (100% males), and BMI (median: 24.05 versus 25.29 kg/m^2^). The Framingham score (median 11 versus 8; *p* = 0.084) and the number of HIV^+^ participants under statin treatment ((12/39; 30.8%) versus (2/22; 9.1%); *p* = 0.064) tended to be superior among TPV^+^ compared to TPV^−^ participants (Table 2). Finally, smoking was more prevalent in TPV^+^ versus TPV^−^ HIV^+^ participants (*p* = 0.005) (Table 2).

In terms of HIV disease parameters, the frequency of HIV^+^ participants with undetectable plasma viral loads was not statistically different in the TPV^+^ compared to the TPV^−^ group (94.8 versus 86.36%, *p* = 0.34). However, TPV^+^ compared to TPV^−^ HIV^+^ participants were on ART for a longer time (mean 16.14 versus 10.69; *p* = 0.003) and the time since infection tended to be longer (median: 19.6 versus 14.7; *p* = 0.06) (Table 2). All 39/39 (100%) TPV^+^HIV^+^ participants received protease inhibitors (PI), while only 14/22 (63.6%) TPV^−^HIV^+^ participants received PI at some point during ART treatment. Nadir CD4 counts were similar in the TPV^+^ and TPV^−^ groups (median 195 versus 170 cells/µL; *p* = 0.397). Regarding laboratory parameters, TPV^+^ and TPV^−^ HIV^+^ participants showed similar white blood cell and lymphocyte counts, as well as D-dimer, LDL, HDL, and triglyceride levels; however, fibrinogen levels, indicative of coagulopathy [59], were significantly increased in TPV^+^ compared to TPV^−^ HIV^+^ (Table 2; *p* = 0.005). Thus, subclinical signs of CVD reflected by the presence of CAA plaques were strongly associated with a longer time on ART, as well as increased plasma fibrinogen levels.

### 3.2. Plasma Markers in Relationship with HIV-1 Status and Subclinical Atherosclerosis

To identify systemic biomarkers associated with subclinical atherosclerosis, we quantified the plasma levels of well-established markers of gut damage (sCD14, LBP, FABP2) and systemic inflammation (CCL20/MIP-3α, MIF, CX3CL1/FKN, CCL25/TECK) in the HIV^−^ and HIV^+^ groups, and in the HIV^+^ group in relationship with subclinical CVD. While higher plasma levels of sCD14, FABP2, CCL20, MIF, and CX3CL1 distinguished HIV^+^ from HIV^−^ participants (Appendix A), only the levels of CCL20, previously associated with HIV disease progression [60], were significantly increased in TPV^+^ compared to TPV^−^ among HIV^+^ participants (*p* = 0.0301) (Appendix A). Thus, plasma CCL20 levels were increased in ART-treated PLWH with subclinical CVD.

### 3.3. T-Cell Profile Alterations in Relationship with HIV-1 Status and Subclinical Atherosclerosis

To study the relationship between key players in adaptive immunity and subclinical CVD status, polychromatic flow cytometry was used to identify specific T-cell subsets in the peripheral blood of HIV^+^ and HIV^−^ participants with/without CAA (Appendix A). Staining with CD45RA and CCR7 was used to identify memory subsets, as previously reported [61]. Surface staining with CD25 and CD127 Abs and intra-nuclear staining with FOXP3 Abs allowed the identification of regulatory T-cells (Tregs; CD127^low^CD25^high^FoxP3^+^ phenotype), as previously reported [61]. Finally, Th17-polarized T-cells were identified among FOXP3^−^ cells by the Boolean combination of gates between CCR6, CD26, and CD161 and defined as cells with a CCR6^+^CD26^+^CD161^+^ phenotype, as previously reported by our group and others [61,62].

HIV^+^ compared to HIV^−^ participants showed decreased frequencies of CD4^+^ T-cells within CD3^+^ T-cells, with increased frequencies of CD8^+^ T-cells and lower CD4/CD8 ratios (Appendix A), consistent with the previous literature identifying the CD4/CD8 ratio as a marker of HIV disease progression [63] and viral reservoirs [64]. No statistically significant differences in the latter parameters were observed among TPV^+^ and TPV^−^ HIV^+^ participants (Appendix A). The frequencies of total memory (CD45RA^−^) CD4^+^ T-cells, as well as those of effector memory (EM, CD45RA^−^CCR7^−^), central memory (CD45RA^−^CCR7^+^; CM), and EM-expressing CD45RA (EMRA) CD4^+^ T-cells, did not differ between HIV^+^ and HIV^−^ participants (Appendix A). Furthermore, there were no significant differences observed between TPV^+^ and TPV^−^ HIV^+^ participants, except for the frequency of EM CD4^+^ T-cells, which was lower in the TPV^+^HIV^+^ group (*p* = 0.03) (Appendix A). Of note, HIV^+^ compared to HIV^−^ participants showed a statistically significant increase in the frequency of Tregs (*p* = 0.0002); similar frequencies of CD4^+^ T-cells with a Th17 phenotype within total memory, EM, and CM subsets; and reduced total memory Th17/Treg (*p* = 0.029), EM Th17/Treg (*p* = 0.007), and CM Th17/Treg (*p* = 0.047) ratios (Appendix A).

Among HIV^+^ participants, TPV^+^ compared to TPV^−^ individuals exhibited similar frequencies of Tregs but decreased frequencies of total memory (*p* = 0.0166), EM (*p* = 0.0083), and CM (*p* = 0.0149) subsets with a Th17 phenotype, and decreased Th17/Treg ratios among memory (*p* = 0.007), EM (*p* = 0.002), and CM (*p* = 0.03) subsets (Figure 1A–C). The same trend was not observed in HIV^−^ individuals, although the low number of HIV^−^ participants available for the study may limit valid interpretations (Appendix A). Thus, ART-treated HIV-1 infection is associated with the expansion of Tregs and an alteration in Th17/Treg ratios, with the paucity of Th17 cells and the subsequent impact on Th17/Treg ratios being exacerbated in HIV^+^ participants with subclinical CAA.

### 3.4. Monocyte Alterations in Relationship with HIV-1 Status and Subclinical Atherosclerosis

Changes in monocyte heterogeneity have been associated with various pathological conditions, including HIV-1 infection [65]. Thus, we investigated changes in monocyte subset frequencies and their phenotypes in relationship with the HIV and TPV status. The flow cytometry gating strategy used to identify monocyte subsets is depicted in Appendix A, as previously described [34,61,66]. An expansion of classical CD14^++^CD16^−^ (*p* = 0.036) to the detriment of intermediate CD14^++^CD16^+^ (*p* = 0.039) monocytes was observed in HIV^+^ compared to HIV^−^, whereas the frequencies of non-classical CD14^−^CD16^++^ and M-DC8^+^ (Slan^+^) monocytes were similar (Appendix A). Within the HIV^+^ group, classical, intermediate, non-classical, and Slan monocytes were similar in frequency regardless of the TPV status (Appendix A).

The phenotype of the monocyte subset was further analyzed, including the expression of the chemokine receptors CCR2 (involved in monocyte trafficking from bone marrow into peripheral tissues [67]), CX3CR1 (involved in intermediate and non-classical monocyte patrolling onto vascular beds [68]), and CCR9 (involved in cell recruitment into multiple sites including the atherosclerotic plaque [69]). Consistent with previous reports [70,71,72], CCR2 and CCR9 were mainly expressed on classical monocytes, while CX3CR1 was predominantly expressed on non-classical and M-DC8^+^ monocytes (Figure 2; Appendix A). The expression of CCR2 and CX3CR1 on monocyte subsets did not differ between the HIV^+^ and HIV^−^ groups (Appendix A). In contrast, the expression of CCR9 on intermediate and non-classical monocytes was significantly diminished in the HIV^+^ compared to the HIV^−^ group (*p* = 0.0087 and *p* = 0.0066, respectively) (Appendix A). A statistically significant increase in the expression of HLA-DR was observed on non-classical (*p* = 0.0258) and Slan/M-DC8^+^ monocytes (*p* = 0.0384) of HIV^+^ compared to HIV^−^ (Appendix A). Further, the stratification based on TPV in HIV^+^ participants demonstrated the similar expression of CCR2 and CX3CR1 on all monocyte subsets, the decreased expression of CCR9 on non-classical monocytes (*p* = 0.0024), and the increased expression of HLA-DR on non-classical (*p* = 0.0081) and Slan/M-DC8^+^ (*p* = 0.0091) monocytes from TPV^+^ compared to TPV^−^ HIV^+^ participants (Figure 2A–D). Thus, in the cohort used for this study, significant alterations were observed in terms of the frequency of classical and intermediate monocyte subsets in relationship with the HIV but not the TPV status; however, changes were observed in the expression of HLA-DR and CCR9, with the predominance of CCR9^low^HLA-DR^high^ non-classical and HLA-DR^high^ M-DC8^+^ monocytes in HIV^+^ versus HIV^−^ participants, a difference further exacerbated within the HIV^+^ group in TPV^+^ versus TPV^−^ participants.

### 3.5. mDC and pDC Frequencies in Relationship with HIV-1 Status and Subclinical Atherosclerosis 

DCs are important immune players acting at the interface between innate and adaptive immunity. While myeloid DCs (mDCs) are mainly involved in antigen presentation and priming the T-cell response [73,74], plasmacytoid DCs (pDCs) are specialized in sensing viruses and producing type I IFN [73]. The flow cytometry gating strategy used to identify mDCs (HLA-DR^+^CD1c^+^) and pDCs (BDCA2^+^CD123^+^) is depicted in Appendix A. Briefly, mDCs and pDCs were gated on CD3^−^CD4^−^ live cells and the expression of CCR6 and CCR9 was also evaluated in each subset. The frequency of mDCs and their expression of CCR6 or CCR9 was similar in HIV^+^ and HIV^−^ participants (Appendix A), with no differences between TPV^+^ and TPV^−^ HIV^+^ participants (Appendix A). Similar results were obtained for the frequency and phenotype of pDCs (Appendix A). Thus, in this cohort, the mDC and pDC frequencies and CCR6/CCR9 phenotype did not distinguish between groups with different HIV and TPV statuses.

### 3.6. Multivariate Analysis Identifies an Immunological Signature Associated with Subclinical Atherosclerosis in ART-Treated PLWH

Logistic regression models were used to investigate the effect of potential confounding factors that might affect the association between the changes in the frequency and/or phenotype of immune subsets and the presence of subclinical CAA plaques (i.e., TPV). Indeed, the univariate logistic regression model with the immune subsets alone (crude association) was compared to multivariate logistic regression models in the presence of four sets of potential confounding factors (Models 1–4), as well as fibrinogen alone (Model 5) (Table 3). In the crude analysis, elevated fibrinogen levels; low Th17/Treg ratios among total memory, EM, and CM CD4^+^ T-cell subsets; and a high frequency of CCR9^low^HLA-DR^high^ non-classical monocytes were associated with the presence of CAA plaques. Conversely, a low frequency of CCR9^+^HLA-DR^low^ non-classical monocytes was associated with the absence of CAA plaques (Table 3). Adjustments were performed for the ART duration, smoking, and stains (Model 1); ART duration, smoking, and LDL (Model 2); ART duration, smoking, and triglycerides (Model 3); and ART duration, HIV duration, and FRS (Model 4), as well as for FRS and fibrinogen separately. The association remained statistically significant for fibrinogen and CCR9^low^HLA-DR^high^ and CCR9^+^HLA-DR^low^ non-classical monocytes in Models 1–4 and for Th17/Treg ratios in Model 4 only (Table 3). Interestingly, M-DC8^+^ monocytes proved to be associated with the presence of TPV only after adjustment in Models 1–4 (Table 3). Finally, after adjustment with fibrinogen alone, the total memory, EM, and CM Th17/Treg ratios remained associated with the presence of plaques, while the frequency of CCR9^+^HLA-DR^low^ non-classical monocytes remained associated with the absence of plaques (Table 3). Together, our studies reveal alterations in Th17/Treg ratios and the non-classical monocyte frequency/phenotype that, together with fibrinogen levels, coincide with subclinical CVD, alterations that may directly/indirectly fuel CAA plaque formation in ART-treated PLWH.

## 4. Discussion

Previous studies by our group and others have documented immunological alterations that occur in PLWH and persist during ART, including gut barrier impairment, microbial translocation, Th17 cell paucity, and monocyte subset expansion/activation (reviewed in [65]). In this study, we investigated whether these alterations were associated with subclinical CAA in ART-treated PLWH in our local Canadian cohort, mainly composed of male participants. To this aim, we had access to plasma and PBMC samples from ART-treated PLWH (HIV^+^) and HIV^−^ participants included in the cardiovascular imaging sub-study of the CHACS [53,54,55]. Markers of gut dysfunction (i.e., sCD14, LBP, I-FABP) and chemokines involved in cell trafficking (i.e., CCL20, CX3CL1, CCL25) were quantified in the plasma, while the frequency and phenotype of CD4^+^ T-cell subsets (i.e., Th17, Tregs) and CD8^+^ T-cells, as well as monocyte (i.e., classical, intermediate, non-classical, M-DC8^+^) and DC (i.e., mDC, pDC) subsets, were monitored in the peripheral blood. Variations in these immunological parameters, together with multiple clinical measurements, were studied in relationship with the presence of CAA plaques visualized/monitored by CT scan, as previously reported [5,55]. Our results support a model in which the paucity of Th17 cells, the alteration of Th17/Treg ratios, and the abundance of CCR9^low^HLA-DR^high^ non-classical monocytes favor a state of systemic immune activation that may fuel CAA in ART-treated PLWH.

Studies previously performed by our group on a total of 181 ART-treated PLWH (HIV^+^) and 84 HIV-uninfected controls (HIV^−^) included in the CHACS at baseline revealed a two- to three-fold increase in the coronary non-calcified plaque burden in ART-treated PLWH compared to HIV^−^ individuals [5]. The current immunological sub-study was performed on a fraction of HIV^+^ (*n* = 61) and HIV^−^ (*n* = 21) participants, the first participants enrolled in the CHACS [53,54,55]. In this sub-study, statistical differences were not observed in the prevalence of CAA plaque detection (TPV > 0) between the HIV^+^ and HIV^−^ groups. Nevertheless, the number of participants with TPV values > 100 mm^3^ was higher among HIV^+^ (32/61; 52.4%; median TPV: 608.7 mm^3^) compared to HIV^−^ participants (7/21; 33.3%; median TPV: 192.8 mm^3^). Consistently, we observed that the levels of HDL were lower, while the levels of triglycerides were higher in ART-treated PLWH compared to HIV^−^ participants. In addition, plasma markers of intestinal damage (FABP2), systemic inflammation (sCD14), and chemokines involved in cell trafficking (CCL20, MIF, and CX3CL1) were significantly higher in HIV^+^ compared to HIV^−^ participants. This is indicative that ART does not restore gut barrier function or counteract systemic immune activation in PLWH, thus explaining the increased CVD risk associated with HIV-1 infection relative to the general population [7,8,10,13,14,15].

The detection of subclinical CAA plaques in HIV^+^, as measured by TPV and LAPV, was associated with a longer time on ART and elevated plasma levels of fibrinogen, as well as a tendency for an increased time since infection, Framingham score, and the use of statins. No differences in nadir CD4 counts were observed. Among the soluble factors monitored in plasma, only levels of the CCR6 binding chemokine CCL20 proved to be higher in TPV^+^ compared to TPV^−^ HIV^+^. CCL20 was previously reported to be increased during HIV infection [60]. Studies in CCR6^−/−^Apo^−/−^ mice demonstrated the role of CCR6 in monocyte recruitment into atherosclerotic plaques, as well as monocyte emigration from bone marrow [54,75]. Of note, monocytes per se can produce CCL20 upon exposure to HIV [76]. Thus, increased plasma levels of CCL20 point to an increased risk of CVD in ART-treated PLWH.

Th17-polarized CD4^+^ T-cells represent key players in immunity at mucosal barriers [77]. Multiple studies have documented alterations in Th17 function and frequency during HIV/SIV infection and identified Th17 depletion as the major cause of gut barrier damage, which results in microbial translocation-induced systemic immune activation [77]. An important finding of our study was the significant reduction in the frequency of Th17 cells with memory, EM, and CM phenotypes in the peripheral blood of TPV^+^ compared to TPV^−^ HIV^+^ participants. The preferential depletion of Th17 cells during HIV-1 infection was linked to their relatively high susceptibility to integrative HIV-1 infection [77]. The loss of Th17 cells during early acute HIV infection correlated with systemic immune activation measured by the proportion of activated CD8^+^ T-cells [78]. Additionally, higher plasma lipopolysaccharide, which is a marker of microbial translocation and gut damage, correlated with lower frequencies of Th17 cells in chronic HIV infection [79]. ART initiation during very early acute HIV infection restored Th17 cells. It was also reported that although long-term ART adherence normalizes the frequency of sigmoid Th17 cells, high levels of plasma lipopolysaccharide were still observed in these individuals [79,80]. In our cohort, we did not observe differences in the frequencies of Th17 cells between HIV^−^ and HIV^+^ participants. This similarity could be explained by the fact that the median ART duration of our participants was relatively high, exactly 15.67 years (IQR: 8.14–19.37 years), raising the possibility that long-term ART may have restored the Th17 compartment in the blood. Nevertheless, the detection of CAA plaques in ART-treated PLWH was associated with a reduced frequency of Th17 cells, which may mirror alterations in mucosal barrier integrity. The existence of a link between alterations occurring at the level of the mucosal intestinal barrier and the pathology of the cardiovascular system supports the existence of the gut–heart axis [10,17,22,31,33,42,45,54,58,80,81,82,83,84,85,86].

The detection of IL-17A-producing CD4^+^ T-cells in the atherosclerotic plaques of mice [2,3,4,6,13,18,20,28,33,77,87,88,89] supports the possibility that subsets of Th17 cells may be recruited into the vascular beds and locally fuel atherosclerosis. In line with this scenario, in our study, TPV^+^ compared to TPV^−^ ART-treated PLWH exhibited increased plasma levels of CCL20, a chemokine essential for CCR6^+^ Th17 cell trafficking [77]. Considering that T-cell function depends on TCR triggering, future studies should determine the antigenic specificity of Th17 cells recruited into atherosclerotic plaques. Indeed, studies have reported that the CD4^+^ T-cells infiltrating the atherosclerotic plaque recognize CMV and HIV peptides, as well as LDL and apolipopotein B peptides [24]. Th17 cells mainly recognize components of the microbiota [90,91]. Meanwhile, specific components of the microbiota were reported to be present in atherosclerotic plaques [92,93], thus supporting a cognate contribution of Th17 cells to the process of atherosclerosis. Nevertheless, the role of Th17 cells may be dual—a positive one at the intestinal level, where they maintain the integrity of the mucosal barrier, and a deleterious one when infected by HIV and recruited into the atherosclerotic plaque and exerting effector functions in response to translocated microbial components.

In contrast to Th17 cells, Tregs are reported to be expanded during HIV-1 infection despite the initiation of ART, with their suppressive functions being detrimental to the proper development of anti-HIV immunity [94]. Due to alterations in the frequency of Th17 and Tregs during HIV-1 infection, the Th17/Treg ratio is decreased during HIV infection [95,96]. In primary HIV-1 infection, the Th17/Treg ratio was negatively correlated with the proportion of activated CD8^+^ T-cells, plasma viral load, and markers of monocyte activation, such as sCD14 and IL-1RA [96]. In line with these results, we observed increased frequencies of Tregs and decreased Th17/Treg ratios in this group of ART-treated PLWH compared to HIV^−^ participants included in the CHACS. Although we did not observe statistically significant differences in the frequency of Tregs between TPV^+^ and TPV^−^ in our group of *n* = 61 ART-treated PLWH, another sub-study performed by our group on *n* = 84 participants from the same CHACS cohort found an increase in Treg frequency in TPV^+^ compared to HIV^+^ participants [27]. Despite this discrepancy, which may be explained by differences in the number of participants included in the two studies, we consistently report here that CAA is associated with a reduction in Th17/Treg ratios in ART-treated PLWH.

Solid literature evidence supports the contribution of specific monocyte subsets to the process of atherosclerosis [34,35], with studies in mice demonstrating the recruitment of non-classical monocytes into the plaque via the chemokine receptor CX3CR1 and its ligand CX3CL1/FKN [97]. Non-classical monocytes indeed express CX3CR1 preferentially and patrol the vascular beds rich in membrane-associated CX3CL1 (reviewed in [65]). Of note, an increased proportion of non-classical monocytes expressing CX3CR1 was positively associated with the coronary intima-media thickness [41,98]. Non-classical monocytes also produce inflammatory cytokines (IL-6), chemokines (CCL2), and MMP-9 upon interaction with CX3CL1-expressing endothelial cells [99]. In addition, the CCL2 receptor CCR2 is expressed on classical and intermediate monocytes and is key in establishing atherosclerosis in mice models [34,35]. In our study, although we did not observe the expansion of intermediate/non-classical CCR2^−^CX3CR1^+^ monocytes, subclinical CAA was associated with the abundance of CCR9^low^HLA-DR^high^ non-classical monocytes. The significance of this correlation was maintained after cofounder factor adjustment and multivariate analyses, thus raising new questions on the role of these monocytes in CVD in ART-treated PLWH.

The expression of CCR9 on monocytes and the CCR9/CCL25 axis have been associated with autoimmune diseases such as inflammatory bowel disease, rheumatoid arthritis (RA), and ankylosing spondylitis [71,100]. In relation to CVD, in a mouse model of atherosclerosis, the presence of monocytes and resident macrophages expressing CCR9 with characteristics of plaque-foaming cells was observed. In addition, CCR9 silencing by RNA interference decreased the size of atherosclerotic plaques [69]. CCR9 is also associated with cardiac hypertrophy in mice models [101]. Furthermore, in humans, CCR9^+^ cells were visualized infiltrating the atherosclerotic plaques [69]. In our cohort, we observed the decreased expression of CCR9 on non-classical monocytes in ART-treated PLWH with subclinical CAA. One of the possible explanations is the infiltration of CCR9^+^ non-classical monocytes into the atherosclerotic plaques, explaining why they are not observed in the periphery. On the contrary, CCR9^+^ pDCs have been reported to be tolerogenic and CCR9 loss to be deleterious in the development of atherosclerosis [50], which suggests that CCR9 is not necessarily a marker of inflammation. In our study, the predominance of CCR9^+^ non-classical monocytes was associated with the absence of CAA plaques. Therefore, one cannot exclude the possibility that CCR9^+^ monocytes play a protective role in atherosclerosis. Changes in CCR9 expression did not correlate with changes in the plasma levels of the CCR9 ligand CCL25. Future mechanistic studies should address this possibility.

Our study has multiple limitations. First, we performed cross-sectional but not longitudinal immunological studies on baseline samples included in the CHACS cohort. This pan-Canadian cohort was initiated in Montréal in 2013 and includes now *n* = 805 HIV^+^ and *n* = 244 HIV^−^ controls for a 5-year longitudinal follow-up [53,54,55]. Participants in the CHACS cardiovascular imaging sub-study (*n* = 181 HIV^+^ and *n* = 84 HIV^−^) are presently undergoing repeat cardiac CT angiograms, which will provide data to inform us on the validity of the identified markers of subclinical CAA in their ability to predict CVD progression in ART-treated PLWH. The current results will orient future longitudinal studies on the CHACS cohort. Second, the CHACS participants are, in the majority, males, and sex-related differences could not be addressed. Indeed, it is reported that differences exist between males and females in terms of HIV and CVD pathogenesis [102], with an increased CVD risk observed in female versus male PLWH [3,88]. Whether the immunological signature that we identified is also associated with subclinical CVD in women remains to be investigated. Third, we did not study alterations in the Th17/Treg ratios in relationship with the size of HIV reservoirs. However, studies performed by our group on the CHACS samples revealed an increased HIV-DNA reservoir in the CD4^+^ T-cells of TPV^+^ versus TPV^−^ HIV^+^ [89], pointing to the possibility that Th17 depletion is a consequence of their infection. Fourth, we were not able to confidently conclude on the immunological differences among HIV^−^ participants with and without subclinical CAA due to the reduced number of individuals enrolled at the time that we performed the study. Nevertheless, despite the small sample size, there was no decrease in Th17 frequency or in Th17/Treg ratios between TPV^+^ and TPV^−^ HIV^−^ participants, indicative that the immunological signature associated with subclinical CAA is likely distinct in HIV^+^ versus HIV^−^ groups.

## 5. Conclusions

In conclusion, we report results generated on peripheral blood cells collected from participants included in the CHACS, a longitudinal cohort established to identify novel biomarker predictors of subclinical CVD in ART-treated PLWH [53,54,55]. Our investigations revealed an immunological signature associated with subclinical CAA in ART-treated PLWH, including (i) increased plasma CCL20 levels (indicative of systemic immune activation); (ii) reduced Th17/Treg ratios (reflecting ongoing alterations in mucosal barrier integrity); (iii) increased frequencies of non-classical CD16^+^ monocytes with a CCR9^low^HLA-DR^high^ phenotype (exerting pro-inflammatory features); and (iv) increased plasma fibrinogen levels (sign of coagulopathy). Based on the immunological differences that we identified between HIV^+^ and HIV^−^ participants with similar subclinical CAA plaque prevalence, we anticipate accelerated CVD progression in the HIV^+^ group. Mechanistic investigations and longitudinal follow-up studies are required to validate whether such immunological alterations indeed fuel CAA plaque development in ART-treated PLWH. Our studies emphasize the need for new therapeutic strategies designed to restore immunological homeostasis in PLWH under long-term ART, especially at mucosal barrier levels, in an effort to reduce the CVD risk.

## Figures and Tables

**Figure 1 cells-13-00157-f001:**
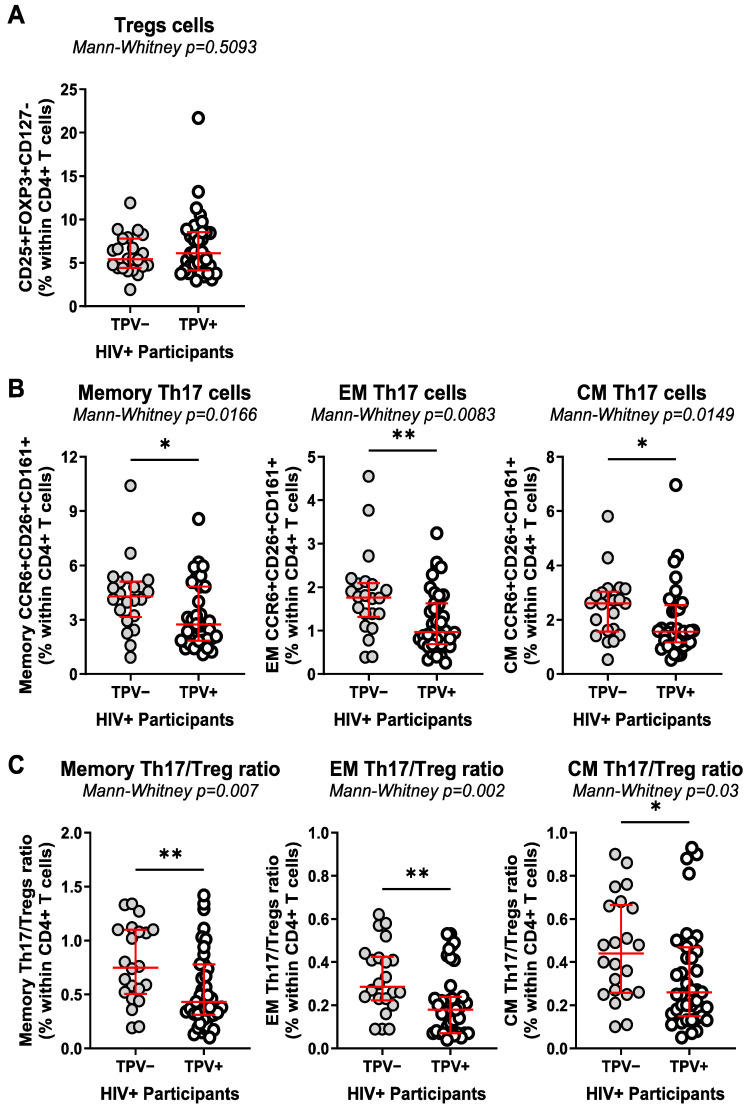
Decreased Th17 frequencies and Th17/Treg ratios coincide with subclinical CAA in ART-treated PLWH. The frequencies of regulatory CD4^+^ T-cells (Tregs; CD25^+^FOXP3^+^CD127^−^CD4^+^) (**A**), as well as Th17-polarized CD4^+^ T-cells (CCR6^+^CD26^+^CD161^+^) with memory (CD45RA^−^), effector memory (EM) (CD45RA^−^CCR7^−^), and central memory (CM) (CD45RA^−^CCR7^+^) phenotypes (**B**), were compared between TPV^+^ (*n* = 39) and TPV^−^ (*n* = 22) HIV^+^ participants. (**C**) Shown are the Th17/Treg ratios within the memory, EM, and CM Th17 subsets among TPV^+^ versus TPV^−^ HIV^+^ participants. Median and IQ range are indicated in red as horizontal lines. Differences among study groups were determined by Mann–Whitney rank test. *p*-values and statistical significance are indicated in the figures (*, *p* < 0.05; **, *p* < 0.01).

**Figure 2 cells-13-00157-f002:**
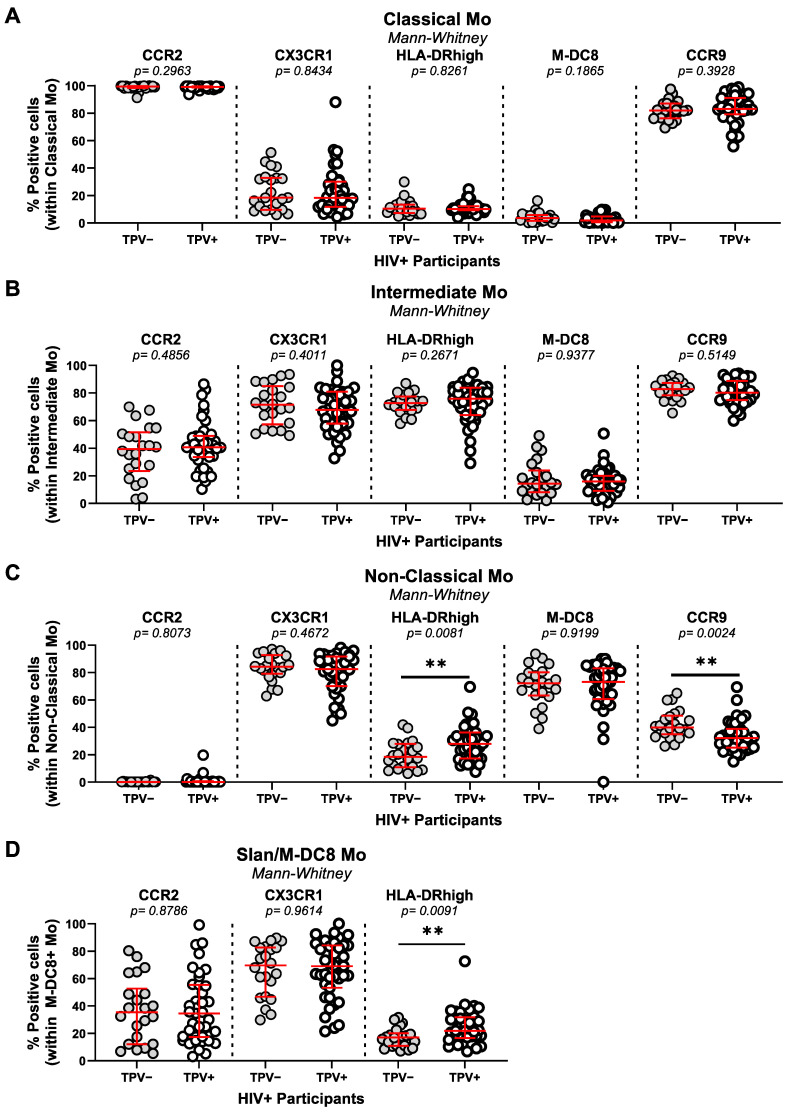
Non-classical CCR9^low^HLA-DR^high^ and M-DC8^+^HLA-DR^high^ monocytes are expanded in ART-treated PLWH with subclinical CAA. Total monocytes (Mo) were identified as cells lacking the T-cell lineage markers CD3 and CD4 and the DC marker CD1c, and expressing HLA-DR (Appendix A). Shown is the expression of CCR2, CX3CR1, CCR9, HLA-DR, and M-DC8 on classical (CD14^++^CD16^−^) (**A**), intermediate (CD14^++^CD16^+^) (**B**), and non-classical (CD14^+^CD16^++^) Mo (**C**), as well as the expression of CCR2, CX3CR1, and HLA-DR on M-DC8+ Mo (**D**), in PBMCs of TPV^+^ (*n* = 39) versus TPV^−^ (*n* = 22) HIV^+^ participants. Median and IQ range are indicated in red as horizontal lines. Differences among study groups were determined by Mann–Whitney rank test. *p*-values and statistical significance are indicated in the figures (**, *p* < 0.01).

**Table 1 cells-13-00157-t001:** Description of study participants.

	HIV^+^ (*n* = 61)	HIV^−^ (*n* = 21)	*p*-Value
Demographics and Clinical Information			
Age (years) *	55.38 ± 6.99	56.94 ± 7.99	0.39
Male &	61 (100%)	17 (81%)	0.003
BMI (kg/m^2^) #	24.43 (22.10–28.29)	25.72 (24.34–28.33)	0.2023
Framingham Risk Score (FRS) #	10 (6–14.25)	8 (8–12.5)	0.67
Current Statin Treatment &	14 (23%)	5 (23.8%)	1
Smoking &			0.502
Never	18 (29.5%)	8 (38.1%)	
Current Smoker	23 (37.7%)	5 (23.8%)	
Former Smoker	20 (32.8%)	8 (38.1%)	
Coronary Artery Disease Parameters			
Total Plaque Volume (TPV; mm^3^) #	108.30 (0–698)	49.70 (0–279.1)	0.32
Low Attenuated Plaque Volume (LAPV; mm^3^) #	34.40 (0–194.5)	10.53 (0–118.9)	0.39
Laboratory Parameters			
White Blood Cells (×10^9^/L) #	5.6 (4.7–7.1)	6.15 (5.65–7)	0.2580
Lymphocytes (×10^9^/L) *	1.90 ± 0.66	1.87 ± 0.47	0.85
LDL (mmol/L) #	2.58 (2.13–3.40)	3.10 (2.30–3.82)	0.21
HDL (mmol/L) #	1.23 (1.07–1.3)	1.44 (1.30–1.6)	0.002
Triglycerides (mmol/L) #	1.73 (1.19–2.99)	1.21 (0.89–1.68)	0.008

HIV^+^, ART-treated PLWH; HIV^−^, uninfected people; NA, not applicable; *, for numeric parametric variables: mean ± SD and *t*-tests were used; # for numeric non-parametric variables: median, interquartile range (IQR), and Mann–Whitney test were used; &, categorical variables were compared with Fisher’s exact test.

**Table 2 cells-13-00157-t002:** Description of ART−treated PLWH with and without subclinical atherosclerosis.

	TPV^+^ (*n* = 39)	TPV^−^ (*n* = 22)	*p*-Value
Demographics and Clinical Parameters			
Age (years) *	56.24 ± 6.35	53.85 ± 7.93	0.1625
Male &	39 (100%)	22 (100%)	NA
BMI (kg/m^2^) #	24.05 (21.05–28.38)	25.29 (22.91–28.27)	0.23
Framingham Risk Score #	11.00 (7–18)	8.00 (5.5–13)	0.085
Current Statin Treatment &	12 (30.8%)	2 (9.1%)	0.064
Smoking &			0.005
Never	7 (17.9%)	11 (50%)	
Current Smoker	20 (51.3%)	3 (13.6%)	
Former Smoker	12 (30.8%)	8 (36.4%)	
Coronary Artery Disease Parameters			
Total Plaque Volume (mm^3^) #	608.70 (142.6–881.8)	0	NA
Low Attenuated Plaque Volume (mm^3^) #	164.80 (36–296)	0	NA
HIV Disease Parameters			
Undetectable Viral Load &	37 (94.8%)	19 (86.36%)	0.34
Duration of HIV (years) #	19.16 (14.29–24.4)	14.71 (4.58–24.14)	0.06
Duration of ART (years) *	16.14 ± 5.68	10.69 ± 7.82	0.003
Antecedent/Current Protease Inhibitor Use &	39 (100%)	14 (63.64%)	<0.0001
Nadir CD4 (×10^9^/L) #	195 (100–330)	170 (78–255)	0.397
Laboratory Parameters			
White Blood Cells (×10^9^/L) #	5.90 (4.7–7.8)	5.40 (4.3–6.85)	0.27
Lymphocytes (×10^9^/L) *	1.91 ± 0.60	1.89 ± 0.78	0.924
D-Dimer (ug/L) #	300.00 (197.5–410)	290.00 (175–380)	0.52
Fibrinogen (G/L) #	3.23 (2.73–3.78)	2.67 (2.27–3)	0.005
LDL (mmol/L) #	2.40 (1.99–3.48)	2.90 (2.23–3.40)	0.35
HDL (mmol/L) #	1.21 (1.07–1.37)	1.25 (0.98–1.37)	0.96
Triglycerides (mmol/L) #	2.07 (1.26–3.1)	1.38 (1.09–2.03)	0.109
Serology			
Co-Infection CMV #	36 (92.3%)	17 (77.3%)	0.39

TPV, total plaque volume; NA, not applicable; *, for numeric parametric variables: mean ± SD and *t*-test were used; #, for numeric non-parametric variables: median, IQR, and Mann–Whitney test were used; &, categorical variables were compared with Chi-square (3 categories) or Fisher’s exact tests (2 categories).

**Table 3 cells-13-00157-t003:** Logistic regression prediction analysis for the association between immune cell subsets and the presence/absence of atherosclerotic plaques in ART-treated PLWH.

Models	OR	Memory Th17/Treg Ratio	EM Th17/Treg Ratio	CM Th17/Treg Ratio	CCR9^+^ HLA-DR^low^ Non-Classical Monocytes	CCR9^low^ HLA-DR^high^ Non-Classical Monocytes	M-DC8+ HLA-DR^high^ Monocytes
Crude Model	OR (CI)	0.49 (0.27–0.86)	0.44 (0.24–0.79)	0.57 (0.33–0.99)	0.30 (0.15–0.63)	3.56 (1.45–8.74)	2.71 (1.24–5.91)
	Adj. p	0.03	0.03	0.05	0.01	0.02	0.10
Model 1	OR (CI)	0.47 (0.22–1.02)	0.43 (0.18–1.03)	0.55 (0.27–1.11)	0.24 (0.09–0.62)	6 (1.77–20.28)	5.39 (1.56–18.6)
	Adj. p	0.1	0.1	0.1	0.015	0.015	0.04
	%Δ OR	3%	2%	4%	22%	70%	98%
Model 2	OR (CI)	0.45 (0.21–0.94)	0.41 (0.18–0.94)	0.52 (0.26–1.03)	0.2 (0.07–0.56)	6.44 (1.89–21.95)	6.06 (1.68–21.93)
	Adj. p	0.07	0.07	0.07	0.01	0.01	0.03
	%Δ OR	9%	6%	9%	35%	80%	123%
Model 3	OR (CI)	0.41 (0.19–0.91)	0.38 (0.16–0.94)	0.48 (0.24–0.97)	0.17 (0.06–0.52)	8.09 (2.06–31.67)	7.13 (1.8–28.22)
	Adj. p	0.06	0.06	0.06	0.01	0.01	0.03
	%Δ OR	17%	13%	16%	44%	126%	162%
Model 4	OR (CI)	0.45 (0.23–0.89)	0.47 (0.24–0.92)	0.49 (0.26–0.94)	0.26 (0.11–0.6)	5 (1.72–14.56)	3.75 (1.4–10.04)
	Adj. p	0.05	0.05	0.05	0.01	0.01	0.05
	%Δ OR	7%	6%	14%	15%	40%	40%
Model 5	OR (CI)	0.45 (0.24–0.86)	0.45 (0.23–0.87)	0.5 (0.27–0.94)	0.26 (0.11–0.62)	3.07 (1.14–8.25)	2.75 (1.12–6.74)
	Adj. p	0.04	0.04	0.04	0.02	0.1	0.15
	%Δ OR	8%	3%	13%	13%	14%	1%

OR, odds ratio; CI, 95% confidence interval; Adj. p, adjusted *p*-values; %Δ (delta) OR, OR difference relative to the crude model; Model 1, adjustment for ART duration, smoking, and statins; Model 2, adjustment for ART duration, smoking, and LDL; Model 3, adjustment for ART duration, smoking, and triglycerides; Model 4, adjustment for ART duration, HIV duration, and Framingham risk score (FRS); Model 5, adjustment for fibrinogen.

## Data Availability

Data from this manuscript are available from the corresponding author upon reasonable request.

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
