# Peer review of "Alterations in Th17 Cells and Non-Classical Monocytes as a Signature of Subclinical Coronary Artery Atherosclerosis during ART-Treated HIV-1 Infection"

_cells, 2024, doi:10.3390/cells13020157_

Round 1

Reviewer 1 Report

Comments and Suggestions for Authors

The authors tried to analyze some factors for coronary artery atherosclerosis (CAA) including ART duration, markers in plasma and immune cells in HIV-infected participants. However, I have some concerns need to be clarified.

1.   There was no difference in the CAA prevalence between HIV(+) and HIV(-) population which may suggest that HIV infection was not related with CAA, at least in this study. So I cannot see the significance of including HIV(-) population in this study.

2.   In this study, there were 61 HIV(+) participants with 39 TPV+ (CAA cases). But in the logistic regression, there were 7 factors (Crude Model), and even more than 7 factors in Model 1-4 (plus adjusted factors). It seems overfitted. Moreover, Memory Th17/Treg ratio, EM Th17/Treg ratio and CM Th17/Treg ratio could not be included simultaneously in the model because Memory Th17 included EM Th17 and CM Th17.

3.   In results 3.6, “In crude analysis, fibrinogen levels, the Th17/Treg ratios among total memory, EM, CM CD4+ T-cell subsets, and the frequency of CCR9lowHLA-DRhigh non-classical monocytes was associated with the presence of CAA plaque, while the frequency of CCR9+HLA-DRlow non classical monocytes was associated with the absence of CAA plaque.” However, when I looked at the OR, fibrinogen levels (2.51), the Th17/Treg ratios among total memory (0.49), EM (0.44), CM (0.57) CD4+ T-cell subsets, and the frequency of CCR9lowHLA-DRhigh non-classical monocytes (3.56). So the description was very confusing. Generally, OR > 1.0 shows positive relationship. Thus, I will think the ratios were negatively related with CAA because their ORs were less than 1.0. Simultaneously, the comparison of ratios in Fig. 1B also showed lower levels in TPV+ than in TPV-.  

4.   Results 3.1 the last paragraph, “subclinical signs of CVD reflected by the presence of CAA plaque were strongly associated with a longer time on ART, as well as increased plasma fibrinogen levels.” In multivariable logistic regression, why only fibrinogen was analyzed as a factor but ART duration as an adjusted factor?  

5.   Fig. S3A, Live cell and T-cells gating marker seems not correct (Y axis).

Author Response

Point-by-point response – CELLS Journal

REVIEWER 1

The authors tried to analyze some factors for coronary artery atherosclerosis (CAA) including ART duration, markers in plasma and immune cells in HIV-infected participants. However, I have some concerns need to be clarified.

RESPONSE : We thank Reviewer 1 for raising the concerns that we addressed here below in an effort to improve the quality of our manuscript.

  1. There was no difference in the CAA prevalence between HIV(+) and HIV(-) population which may suggest that HIV infection was not related with CAA, at least in this study. So I cannot see the significance of including HIV(-) population in this study.

RESPONSE: We agree with the observation made by Reviewer 1 that the 2 groups of HIV+ and HIV- participants are not different in terms of CAA prevalence, as illustrated in Supplemental Figure 1A. We also acknowledge the fact that we did not have enough statistical power to capture this difference given the limited number of HIV- participants. Yet, this might be explained by the fact that HIV+ and HIV- participants enrolled in the CHACS presented without clinical signs of CVD and with a similar Framingham score (median 8 and IQR 8-12.5 for HIV- participants; median 10 and IQR 6-14.25 for the HIV+ participants; p-value: 0.67; see Table 2). The goal of such enrollment was to capture immunological correlates of CVD in PLWH that are independent of the traditional risk factors. Nevertheless, as stated in the 2nd paragraph of the discussion, “the number of participants with TPV values >100 mm3 was higher among HIV+ (32/61; 52.4%; median TPV: 608.7 mm3) compared to HIV- participants (7/21; 33.3%; median TPV: 192.8 mm3)”. Importantly, statistically significant differences were also observed between the HIV+ and HIV- groups in terms of plasma levels of markers of systemic inflammation (sCD14, FABP2, CCL20, MIF, CXC3CL1) (Suppl. Figure 2), indicative of an increased state of systemic inflammation in HIV+ and HIV- participants.

In response to this criticism, the manuscript was revised to clarify the fact that the scope of this cohort was to recruit at baseline HIV+ and HIV- participants with similar subclinical CAA prevalence for a longitudinal follow-up. Based on differences we identified, we anticipate a more rapid CVD progression in HIV+ and HIV- participants.

  1. In this study, there were 61 HIV(+) participants with 39 TPV+ (CAA cases). But in the logistic regression, there were 7 factors (Crude Model), and even more than 7 factors in Model 1-4 (plus adjusted factors). It seems overfitted. Moreover, Memory Th17/Treg ratio, EM Th17/Treg ratio and CM Th17/Treg ratio could not be included simultaneously in the model because Memory Th17 included EM Th17 and CM Th17.

RESPONSE: In the logistic regression model each immune subset (factor) was tested alone (the crude univariate model) independently of the others. The results of the crude model are shown in Table 3 (Crude model). To test for the effect of other confounder variables, we used a multivariate logistic regression in which we tested each immune factor but in the presence of a set of other potential clinical variable (Models1-4). The number of variables in each model was kept small to avoid overfitting given our sample size. Also, the immune factors were never tested simultaneously in any model. For the purpose of clarity, the beginning of section 3.6 has been revised.

  1. In results 3.6, “In crude analysis, fibrinogen levels, the Th17/Treg ratios among total memory, EM, CM CD4+ T-cell subsets, and the frequency of CCR9lowHLA-DRhighnon-classical monocytes was associated with the presence of CAA plaque, while the frequency of CCR9+HLA-DRlow non classical monocytes was associated with the absence of CAA plaque.” However, when I looked at the OR, fibrinogen levels (2.51), the Th17/Treg ratios among total memory (0.49), EM (0.44), CM (0.57) CD4+ T-cell subsets, and the frequency of CCR9lowHLA-DRhigh non-classical monocytes (3.56). So the description was very confusing. Generally, OR > 1.0 shows positive relationship. Thus, I will think the ratios were negatively related with CAA because their ORs were less than 1.0. Simultaneously, the comparison of ratios in Fig. 1B also showed lower levels in TPV+ than in TPV-.

RESPONSE: We agree with the reviewer that the description was not clear. The text was modified to indicate the direction of the comparisons as follows:

A logistic regression model was used to determine the association between covariates and the presence or the absence of subclinical CAA plaque (i.e., TPV). In the crude analysis, elevated fibrinogen levels, low Th17/Treg ratios among total memory, EM, and CM CD4+ T-cell subsets, and a high frequency of CCR9lowHLA-DRhigh non-classical monocytes were associated with the presence of CAA plaque. Conversely, a low frequency of CCR9+HLA-DRlow non-classical monocytes was associated with the absence of CAA plaque.”

  1. Results 3.1 the last paragraph, “subclinical signs of CVD reflected by the presence of CAA plaque were strongly associated with a longer time on ART, as well as increased plasma fibrinogen levels.” In multivariable logistic regression, why only fibrinogen was analyzed as a factor but ART duration as an adjusted factor?

RESPONSE: We agree with the reviewer and understand his concerns regarding this point. The use of fibrinogen, as outcome, in the logistic regression was confusing and misleading. The association of ART duration and plasma fibrinogen levels with the outcome was studied using a non-parametric Wilcoxon test and the results of this association are presented in Section 3.1 (Tables 1-2). However, in a distinct analysis the confounding effect of fibrinogen and ART duration were tested separately in two different models using the multivariate logistic regression modeling presented in Section 3.6 (Table 3). To avoid any confusion, Table 3 was revised by removing the column fibrinogen keeping only the results of the confounding effect analysis using ART duration in Models 1-4 and fibrinogen in Model 5.

  1. S3A, Live cell and T-cells gating marker seems not correct (Y axis).

RESPONSE: In response to this criticism, we clarify that the titles of the dot plots represent the events analyzed from the preceding gating. Additionally, we have corrected the y-axis of the Live cells plot and replaced “CD4 PerCP-Cy5.5” with “SSC-A”. In the T-cell plot, we have rectified “CD4+ T cells with “CD4- T cells”.

Reviewer 2 Report

Comments and Suggestions for Authors

Great work.

Would it be possible to add more details about the cART received? In particular, how many years on PIs among TPV+ and TPV- subjects?

Author Response

Point-by-point response – CELLS Journal

REVIEWER 2

Great work.

RESPONSE: We thank Reviewer 2 for the appreciation of our manuscript

  1. Would it be possible to add more details about the cART received? In particular, how many years on PIs among TPV+ and TPV- subjects?

RESPONSE: We obtained information about whether participants had received protease inhibitor use as part of their antecedent or concurrent treatment during the study. In the TPV+ group, all 39 participants (100%) received protease inhibitors, while in the TPV- group, 14 out of 22 participants (63.64%) received protease inhibitors at some point during ART treatment. This additional information has been incorporated into revised Table 2 and is also added to revised Section 3.1, as follows:

In terms of HIV disease parameters, the frequency of HIV+ participants with undetectable plasma viral loads was not statistically different in TPV+ compared to TPV- groups (94.8 versus 86.36%, p=0.34). However, TPV+ compared to TPV- HIV+ participants were on ART for a longer time (mean 16.14 versus 10.69; p=0.003), and the time since infection tended to be superior (median: 19.6 versus 14.7; p=0.06) (Table 2). All 39 (100%) TPV+ participants received protease inhibitors (PI), while only 14 (63.64%) TPV- participants received PI at some point during ART treatment”.

Round 2

Reviewer 1 Report

Comments and Suggestions for Authors

The manuscript has been greatly improved. However, one small thing needs to be clarified. The title of Table 3: fibrinogen should be deleted? (because the authors have deleted the fibrinogen analysis); accordingly, the text line 372 "elevated fibrinogen levels" also needs to be deleted?

Author Response

We thank the Reviewer 1 for the appreciaiton of our revised manuscript. We now revised the title of Table 3 as propertly pointed by Reviewer 1.